# Bacterial Infection in the Sickle Cell Population: Development and Enabling Factors

**DOI:** 10.3390/microorganisms11040859

**Published:** 2023-03-28

**Authors:** Lucrèce M. Délicat-Loembet, Mohamed Ag Baraïka, Flabou Bougoudogo, Dapa A. Diallo

**Affiliations:** 1Department of Biology, Faculty of Sciences, University of Sciences and Techniques of Masuku (USTM), Franceville BP 901, Gabon; 2NGO Sickle Cell Disease Organization of Gabon, Moanda 27/28, Gabon; 3Sickle Cell Disease Research and Control Center (CRLD), Bamako 03 BP 186, Mali; 4National Institute of Public Health (INSP), Bamako, Mali; 5Faculty of Pharmacy, University of Sciences, Techniques and Technologies of Bamako (USTTB), Bamako BP E2528, Mali

**Keywords:** bacterial infections, sickle cell anemia, risk factors

## Abstract

The high frequency of bacterial infections represents a major threat to public health. In developing countries, they are still responsible for significant morbidity and mortality in pediatric populations with sickle cell disease, particularly in children under 5 years of age. Indeed, they have an increased susceptibility to bacterial infections due to their immune deficiency. This susceptibility is even greater for pneumococcal and salmonella infections. In addition, the underdevelopment of some countries and socio-economic factors increases this condition. This review examines the common and specific factors leading to infections in people with sickle cell disease in different types of developed and undeveloped countries. The threat of bacterial infections, particularly those caused by S. pneumoniae and Salmonella, is of increasing concern due to the rise in bacterial resistance to antibiotics. In light of this disturbing data, new strategies to control and prevent these infections are needed. Solutions could be systematic penicillin therapy, vaccinations, and probabilistic antibiotic therapy protocols.

## 1. Introduction

Sickle cell anemia is the most serious form of inherited hemoglobin disease [1], and it represents a major public health problem worldwide. Africa is the most affected continent, with a prevalence of sickle cell gene carriers ranging from 2 to 30%, depending on the geographical area [1,2,3,4]. The natural history of the sickle cell anemia is characterized by chronic anemia that occurs early in life, and it is accompanied by simple or complicated vaso-occlusive crises (VOCs), which can sometimes jeopardize a patient’s vital prognosis and cause chronic, life-threatening, and/or functional complications [5,6,7,8]. Individuals with sickle cell anemia have a high frequency of infection, and this is especially the case in children of less than five years of age. In children with sickle cell anemia, coughing accompanied by fever in those less than 5 years of age and fever with osteoarticular pain in older children should be considered to be a suspected bacterial infection [9]. The complications arising from these infectious diseases are responsible for high rates of mortality, often involving the vital prognosis of sickle cell disease [7,8,10]. The high susceptibility of children with sickle cell anemia to bacterial infections is known to be due to functional asplenia resulting from repeated spleen infarctions [8,11,12], associated with genetic, immunological, and sometimes environmental factors [13,14,15]. All of these may contribute to this high susceptibility to infections in individuals with sickle cell anemia. This review aimed to provide an update on factors that promote and exacerbate bacterial infections in a population with sickle cell anemia and to propose some directions for a better control of infections in sickle cell anemia patients. We will highlight during the work several factors favoring pneumococcal and salmonella infections, such as immunodeficiency, underdevelopment, emergence of antibiotic resistance, while exposing on the epidemiology, evolution, prevention, and treatment of bacterial infections in the person living with sickle cell disease.

## 2. Materials and Methods

After having listed the different factors that are most often observed associated with bacterial infections in sickle cell patients, we proceeded with a targeted search on different research platforms databases. First, we conducted a preliminary literature search in order to evaluate the existence (in number and type) of publications concerning the chosen theme. In this way, we were able to define whether the chosen axes had already been developed previously. To conduct this work, we have using documentary research in Medline PubMed, Google scholar, Google, and Mendeley electronic databases. Key words, such as bacterial infection, sickle cell disease, sickle cell disease, susceptibility to infections, bacterial diversity in sickle cell patients..., were used, and after reading the abstract or full text, we retained the articles that best fit the objectives of our work. This selection work has been carried out for more than 6 months, i.e., throughout the writing of this review. The abstracts or full text of articles that provide information on the subject matter were included and have given to the subdivision of the different sections or parts developed in the heading of the results.

## 3. Results

### 3.1. Immune Deficiency, a Factor That Promotes Pneumococcal and Salmonella Infections

A large number of previous studies have described the natural history of infections in pathogen-dominated sickle cell disease. However, the epidemiological situation in developed countries has changed considerably in recent times. By contrast, genetic and immunological factors can lead to serious life-course infections in sickle cell anemia [1,2,3].

From a genetic perspective, although sickle cell disease involves the same genetic mutation, some patients become disabled by frequent crises and long-term complications, while others, by contrast, can lead an almost normal life. Thus, sickle cell patients appear to be differentially predisposed to specific pathological (clinical or infectious) manifestations of the disease [16]. Polymorphisms in a number of genes involved in the immune response are thought to contribute to the increased susceptibility of individuals with sickle cell anemia to infection [17].

The high susceptibility of individuals with sickle cell anemia to pneumococcal infections is not thought to be linked to any inherent defect in the body’s various defenses against pathogens as the functional capabilities of polymorphonuclears are essentially normal [18]; the humoral immune response following vaccination is normal [19], and the production of immune factors is normal albeit reduced when the stimulus is introduced intravenously [18]. Two anomalies dominate the physiopathology of pneumococcal infections in sickle cell patients: the decrease in the opsonizing power of serum (OPS) and functional asplenia. The decrease in the OPS in sickle cell anemia has been extensively documented with pneumococcal disease [20,21]. However, the reason for this is not entirely clear. Numerous authors have described an anomaly in the alternative complement pathway (ACP) [22]. This anomaly could be due to a decrease in factor B production, itself related to excessive activation of the ACP secondary to hemolysis, for example. This decrease in factor B production has not been confirmed by all authors [20]; changes in the OPS are similar during both crisis and asymptomatic periods [19], thus suggesting that they are not necessarily related to the underlying hematological phenomena. However, this does not rule out the existence of a subgroup of sickle cell anemia with a decrease in OPS in relation to pneumococcal infection, and this would expose these individuals more than those with a normal opsonic activity of serum [23]. Some bacterial pathogens are directly recognized by macrophages, while others require previous opsonization, which is the coating of the microbial surface by complement components (C3b) or other molecules that interact with phagocyte receptors.

The spleen is the primary site for the synthesis of tuftsin, which is an immunostimulatory peptide that participates in activation of the complement system [24]. During early childhood, many major infections can occur while the spleen is still only partially functional, with an increased risk of persistence despite modern prophylactic measures; this phenomenon suggests the presence of additional immune disorders [17]. The spleen plays a key role in the increased susceptibility of individuals with sickle cell anemia to certain bacterial infections, as it functions as a phagocyte filter. Children with sickle cell anemia, who frequently suffer from splenomegaly, are more likely to be infected with pneumococcal disease, which often occurs in early childhood [25]. The reason for this is functional asplenia, highlighted by the absence of an isotopic fixation [26]. This can be reversed by blood transfusion, a bone marrow transplant, or hydroxyurea treatment in some young children [24,27]. This splenic dysfunction leads to the failure to eliminate vesiculated/pocked red blood cells, which by splenic scintigraphy has been correlated with a proportion of vacuolated red cells of more than 3.5% [28]. Pearson et al., using this hematological theory, were able to demonstrate early onset of functional asplenia in individuals with homozygous SS sickle cell disease before 2 years of age [28]. This is observed when the HbF rate decreases to less than 15%, confirming the theory that patients with high HbF levels have a less severe lifetime course of the disease [29]. In the long run, repeated occurrences of vaso-occlusive crises and ischemic lesions associated with progressive sclerosis of the small-diameter precapillary arteries lead to infarction of the spleen tissue. Unable to regenerate, the spleen becomes scarred and atrophied, thereby leading to autosplenectomy, with a narrowing of the spleen resulting in altered splenic function [30].

In addition to a high susceptibility to pneumococcal infections, the susceptibility of patients with sickle cell anemia to salmonella infections has led to a considerable body of scientific research [31]. Opsonized bacteria are eliminated by spleen and liver macrophages, but poorly opsonized bacteria are essentially only eliminated in the spleen. This mechanism generally involves bacteria that are encapsulated, whereby the polysaccharide-like capsules prevent the bacterial wall from interacting with macrophage receptors.

Macrophages represent the main form of cell-mediated immunity against Salmonella. Intestinal macrophages phagocytose Salmonella bacteria that have crossed the coecum mucosa barrier at the level of the Peyer’s patches. Inactivated macrophages are unable to eliminate bacteria that are too numerous or too virulent, which may then migrate to the mesenteric ganglia and the liver, where they can multiply and disseminate. Activation of the ACP by the PSL (pathogen-specific lymphocytes) occurs in the absence of specific immunity. The acquisition of specific immunity enables the activation of macrophages and the destruction of bacterial cells. Capillary occlusion, reduced immunity, and genetic and immunogenetic factors contribute to colonization and proliferation by Salmonella in patients with sickle cell disease [14,15]. Several factors may explain the frequency of severe salmonellosis in patients with sickle cell anemia, such as the saturation of the macrophagic system associated with erythrophagocytosis [6] and, by contrast, the defect of the OPS against Salmonella that can be an aggravating factor [32]. The frequency of Salmonella bone localization is probably related to local factors corresponding to areas of bone infarction.

### 3.2. Underdevelopment as a Factor in Pneumococcal and Salmonella Infections

*Pneumococcus* bacteria can be responsible for septicemia, meningitis, and pneumonia; while Salmonella can cause septicemia, osteomyelitis, osteoarthritis, and cholecystitis [9]. While the prevention of pneumococcal infections is systematic in developed countries, it is lacking in Africa because there are two main obstacles to the prevention of pneumococcal infections. First and foremost, the cost of treatment is a major obstacle due to the very high number of extremely poor families. Diagne et al. showed that only 21% of sickle cell patients treated at the Albert-Royer Hospital in Senegal had been vaccinated against pneumococcal disease [33]. It strikes us that the implementation of an effective preventive policy needs to be achieved not only by pharmaceutical companies providing vaccines at cost but also through the adoption of preventive measures by the governments involved, such as the inclusion of pneumococcal vaccines in the WHO Expanded Vaccination Program (EVP) and establishment of a center for the development of care programs, as has been performed to great effect in Cotonou, Benin [34]. Another important element in achieving pneumococcal prevention is being able to overcome the distrust of a significant number of patients and their families who might otherwise prefer to use traditional medicine [35]. Other factors include malnutrition, a lack of clean drinking water, poor sanitary conditions that increase the risk of bacterial infections, and the emergence of multidrug-resistant bacteria in this vulnerable population [36]. The lack of basic sanitary practices is responsible for salmonella infections in the general population [36]. Another source of salmonella infection is the ingestion of contaminated eggs, meat, raw milk, or vegetables, or contact with an asymptomatic carrier or animal. These factors are major determinants of health. Indeed, it is widely recognized that poverty is associated with poor health [37]. This is particularly relevant for sickle cell anemia, which is the most common genetic disease in poor countries [38].

### 3.3. The Emergence of Antibiotic Resistance, a Factor That Favors Bacterial Infections

The main factors that promote the spread of antibiotic resistance in sickle cell disease are the international recommendations for the systematic prescription of broad-spectrum antibiotics in the event of fever, particularly in individuals with sickle cell anemia [13,39,40]. We know that these international recommendations can lead to the misuse and inadequate use of broad-spectrum antibiotics. The use of antibiotics by the agri-food industry has led the World Health Organization (WHO) to express concern regarding the emergence of resistant strains. Cross-resistance to ampicillin, sulfonamides, streptomycin, chloramphenicol, and tetracycline has emerged, resulting in the recommendation to use fluoroquinolones. Reference center of sickle cell disease in Mali that follows this recommendation, a study assessing the sensitivity of bacteria to antibiotics has revealed a high rate (36.13%) of ESBL-producing enterobacteriaceae [41]. The continued increase in the frequency of ESBL-producing strains is likely to be linked in our study population to the international recommendations. The seriousness of these infections justifies prevention efforts in these patients with sickle cell anemia [42]. The current development of penicillin-resistant pneumococcal resistance is a reason for concern, and it requires effective antibiotic treatment for a possible pneumococcal resistance with an acute febrile picture or meningitis [13]. Over the past year, we have seen more and more C3G-resistant strains of different serotypes; C3G is a key antibiotic drug in pediatric treatments [42,43]. The observed resistance mechanism is due to extended-spectrum beta-lactamases, plasma cephalosporinases, and more recently, carbapenemases [42,44,45]. The spread of resistance to C3G has been reported to be due to their use in the animal sector and due to the importation of strains from Africa through adopted children [44].

### 3.4. Evolution of Bacterial Infections in the Time of Sickle Cell Disease

A number of studies worldwide have provided an up-to-date picture of antibacterial prophylaxis for bacterial infections in sickle cell disease [44,45,46,47,48]. These studies show a gradual decrease in the incidence of these bacterial infections during sickle cell infections, especially in developing countries as well as in the USA and Europe, where vaccination has become a reality in the management of this disease.

Studies in malaria-endemic developing countries have shown that encapsulated bacteria do not make a significant contribution to mortality and morbidity in children with sickle cell anemia and that these children are more likely to succumb to infections by other pathogens [49].

The frequency not only varies with the age of the child but also with the region and the time of study. Salmonella has gradually been overtaken by other organisms, such as Klebsiella [49,50,51], *Pneumococci* [52,53], and *Staphylococcus aureus* [54]. The mechanisms of high susceptibility of sickle cell anemia were developed for an initial bacterial ecology based on *Salmonella* spp. and *S. pneumoniae* found in this population. Recent studies have revealed a different ecology than what was known a few years ago [55,56]. It will be important to study the susceptibility mechanisms associated with the new bacterial ecology.

### 3.5. Bacterial Infections in Pediatric Populations with Sickle Cell Anemia

Infections are one of the most common complications of sickle cell anemia. Infections and infectious complications are the leading cause of morbidity and mortality among patients with sickle cell anemia, especially in those less than five years of age. These infections are a common cause of hospitalization and vaso-occlusive crisis [29]. They occur throughout the life of individuals with sickle cell anemia, and they can put their life at risk, especially in infants and young children. This high susceptibility to infection is observed mainly in homozygous sickle-cell or S/β0-thalassemia but also, albeit to a lesser degree, in other major sickle cell syndromes. Bacterial infections are capable of rapid spread and severe localizations. In infants, even a common viral infection can suddenly trigger acute or subacute spleen sequestration. These infections are also responsible for vaso-occlusive crises via factors that trigger vaso-occlusive crises of sickle cell anemia, such as fever, hypoxia, and dehydration. As a result, a vicious circle is often created between infection and the sickle cell crisis. The most common acute complications encountered during childhood are intense painful crises, serious infections, meningitis, septicemia, and osteomyelitis. Deaths in this age group are secondary outcomes of deadly infections, spleen sequestration, or life-threatening crises [57].

### 3.6. Epidemiology of Bacterial Infections in Individuals with Sickle Cell Anemia

#### 3.6.1. Sickle Cell Anemia and Bacteremia due to Pneumococcal and Salmonella Infections

Pneumococcal disease is the top priority in all scientific studies where it has been isolated, and according to Barrett Connor, the risk of *S. pneumoniae meningitis* is 5–100 times higher in individuals with sickle cell disease than in the general population [58]. The study in the United States in 1986, reported 178 bacterial diseases in a sickle-cell population of 3451 individuals over a 24-month period. *S. pneumoniae* was the most frequently isolated pathogen, accounting for 67% of cases in children six years of age and only 19% of cases in those over six years of age. *Salmonella* was the third most frequently isolated organism, and it was associated with osteomyelitis in 77% of cases. *Haemophilus influenzae* was the third most frequently isolated infectious agent from hemocultures. This study was conducted between 1979 and 1981, before systematic pneumococcal prophylaxis [58,59]. *Salmonella* is isolated most frequently in tropical areas, due to the high frequency of salmonella infections in these regions as a result of poor hygiene [60]. In a study conducted in Gabon in 1999 that aimed to shed light on the different clinical aspects of bacterial infections in children with sickle cell disease in Libreville and to optimize their therapeutic care, the authors reported that *Salmonella* was the most frequently isolated organism from hemocultures, followed by *Streptococcus pneumonia* [61].

*Salmonella* was the second most frequently isolated organism in bacteremia in a sickle cell population in Kenya, after *Pneumococcus*, and the third most frequently isolated in England and the USA, being found in 20% to 50% of bacteremias in these patients [47,58,62].

Despite these diagnostic difficulties, certain complications of sickle cell anemia and severe infections must be distinguished, in particular pneumopathy and acute thoracic syndrome, osteomyelitis and bone infarction, and ultimately sepsis and multifocal crisis [47,58,63].

The severity of these infections justifies the preventive efforts made in these patients. The current development of pneumococcal resistance to penicillin raises concerns and requires the implementation of effective antibiotic treatment to avoid possible pneumococcal resistance when treating acute febrile symptoms or meningitis [13]. However, the severity of a prognosis is not linked to the resistance of pneumococci alone but to the fulminating nature of this type of infection in sickle cell anemia, especially in the first two years of life [13].

Salmonella septicemias are often isolated or associated with osteomyelitis. They usually occur before 10 years of age. All *Salmonella* serotypes are possible, but in a recent publication by Wright et al., *S. enteritidis* was involved in 36% of cases [13].

#### 3.6.2. Sickle Cell Anemia and Invasive Bacteremias or Infections

In an American multicenter study of a population of children with pronounced sickle cell syndrome SS and SC sickle cell disease, reported 178 episodes of bacteremia in 3451 patients followed-up over two years. During the first years of life, the incidence of bacteremia was the same for both the SS and SC forms. The causative organism was not found in 40% of cases of bacteremia; the remainder of the cases mainly involved pneumonia and meningitis. In this study, infants and children under five years of age were found to be more susceptible to pneumococci than older children. There were 18 deaths among the 983 SS homozygous sickle cell patients and 2 among the 277 SC sickle cell patients; pneumococcal meningitis was responsible for the very high (24%) mortality rate in these infections before three years of age [58]. In the 1999 Gabon study cited above, the authors reported that *Salmonella* spp. was the most frequently isolated organism from blood cultures, followed by *Streptococcus pneumoniae*. Urinary tract infections, mainly caused by *E. coli*, *K. pneumoniae*, or *Enterobacter cloacae*, were found in 42.8% of cases [61]. In the Malian study, the main bacterial strains isolated in blood and/or urines are, respectively, *E. coli*, *K. pneumoniae*, or *Salmonella* group [41].

A 2009 study in Kenya in a sickle cell population reported bacteremia due to *Streptococcus pneumoniae*, *Salmonella* non-typhoid, *Haemophilus influenzae*, and *E. coli* in 41%, 18%, 12%, and 7% of cases, respectively [62]. However, a prospective study in Jamaica [44] to investigate the significance of fever in a population of 144 individual with homozygous SS sickle cell disease reported bacteremia in 6.1% of cases and urinary infections in 2.4% of cases, mainly due to *E. Coli*. In 2014, a retrospectively identified the main causes of fever in 108 children with sickle cell disease in Libreville, Gabon, out of 118 admissions for fever. Urinary infections were found in 3.9% of patients, and invasive infections were found in 8.7% of patients. The bacteria isolated from these infections were *E. coli*, *K. pneumoniae*, *Streptococcus pneumoniae*, and *Salmonella* spp. [8]. In this Gabonese study, it was reported that fever was associated with plasma infection in 43.5% and 55.25% of cases of fever without known etiologies [56,64]. An association of bacteremia with low hemoglobin levels was revealed in a study in Tanzania. In these bacteremias, *Staphylococcus aureus*, *Streptococcus pneumoniae*, and *Salmonella typhi* were implicated in 28%, 7%, and 5% of cases, respectively [65].

In Mali, we found a 4.32% rate of bacteremia in children infected with febrile sickle cell disease. In addition to these infections, we found an association of urinary or invasive infection in significant proportions with simple VOCs, but also with life-threatening complications, such as acute chest syndrome, severe anemia, and splenic sequestration, as well as simple symptoms, such as a cough; angina; headache; and abdominal, shoulder blade, chest, and joint pain [41].

#### 3.6.3. Sickle Cell Anemia and Bacterial Respiratory Infections

Pulmonary complications are major determinants of sickle cell morbidity and mortality. It is often difficult to distinguish between simple pneumopathy and the onset of acute chest syndrome in case of a combination of a fever and a radiological outbreak. Acute chest syndrome (ACS) is an acute lung complication characterized by a combination of chest pain with dyspnea and a recent radiological abnormality. The main mechanisms identified are alveolar hypoventilation; pulmonary artery thrombosis is less common [66]. These different causes are often associated. In 2000, a multicenter study in the US of 671 cases of ACS showed the presence of infectious agents in 249 of the cases. The causative agents included *Chlamydia pneumoniae*, *Mycoplasma hominis*, and parvovirus B19, known also as EB19 [67,68]. In this study, 18 deaths were reported. The identified pathogens were *Streptococcus pneumoniae*, *E. coli*, *H. influenzae*, *Legionella*, cytomegalovirus, *Staphylococcus aureus*, and *Chlamydia pneumoniae*. The atypical bacteria *Chlamydia pneumoniae* and *Mycoplasma pneumoniae* appear to be involved in many cases of ACS, the latter especially in young children. These organisms could not be detected until recently due to technical limitations. The severity of these infections is greater in patients with sickle cell anemia than in healthy individuals [69,70]. Another study in Jamaica showed that 35.2% of cases of ACS were associated with bacterial infections.

#### 3.6.4. Sickle Cell Anemia and Osteomyelitis

Osteomyelitis is a classic complication, and it is known for its severity in children with sickle cell anemia. However, its frequency is very different nowadays due to factors, such as the lack of hygiene in developing countries and its rarity in industrialized countries, which means it tends to be confused with bone infarction. The first case, in 1951, was described as being linked with infection due to *Salmonella*, and osteoarticular locations have been frequently reported [71,72]. A prospective study conducted by Richards et al. showed that 86% of non-typhoid *Salmonella* septicemias in sickle cell patients were associated with bone involvement [72].

Salmonella bacteria are the most frequently responsible bacteria: 74% in a 1981 series by Givner et al. in the United States [73], 50% of 14 episodes noted in an American study over 27 years [74], and 88% of osteomyelitis cases reported by Omanga in a study carried out in the present-day Zaire Democratic Republic of Congo [75]. *Staphylococcus aureus* is rarely found. Osteomyelitis is often found at multiple sites, and it can reach atypical bones, such as the small carpal bones, the metacarpals, the phalanges, the vertebrae, and the sternum. Pathological fractures are possible. These locations and their multiplicity are explained by the fact that osteomyelitis almost always occurs in the aftermath of a complicated vaso-occlusive crisis of bone infarction.

Differential diagnosis between bone infarction (itself to be differentiated from a simple vaso-occlusive crisis) and acute osteomyelitis is very difficult, since in both cases, during the vaso-occlusive crisis, bone pain, fever, and biological inflammatory syndrome are observed. In France, where the frequency of bone infarction is more than ten times higher than that of osteomyelitis, technetium scintigraphy appears to be the most discriminating test in our experience, provided it is employed very early, as reported for two cases of sickle-cell children who had osteomyelitis due to *Salmonella* [76]. In the United States, 67% of Salmonella isolated during invasive infections were responsible for signs of osteomyelitis in sickle cell children [71]. In a study carried out in Mali at the Gabriel Touré University Center in 2015, the authors reported three cases of osteoarticular infections due to *Salmonella enterica* serotype Typhi [74].

### 3.7. Prevention and Treatment of Fatal Bacterial Infections in Sickle Cell Anemia

Invasive pneumococcal infections were analyzed 20 years ago in the United States, and they were found to be the leading cause of death in children with sickle cell disease [77]. In 1986, a randomized study showed that a daily penicillin treatment given twice a day reduced the incidence of pneumococcal infections by 84% compared to the placebo-treated group [78]. The reduction in mortality in infants treated early with penicillin following neonatal screening led to the detection of sickle cell anemia at birth in many northern countries.

Given the risk of non-compliance and the increase in the frequency of strains with reduced sensitivity to penicillin, pneumococcal vaccination should also be systematically carried out in children with sickle cell anemia. For a long time, Pneumo23^®^ polysaccharide was the only vaccine available, effective after the age of 2 years, which is now routinely preceded in the United States and Europe by a seven-valent conjugate vaccine, Prevenar^®^, administered at 2, 3, and 4 months, with a booster in the second year. A recent review from Uganda discussed the merits of preventing pneumococcal disease [45]. Nevertheless, publications in West and Central Africa strongly advocate the application of the prevention of pneumococcal infections in children with sickle cell anemia. The prevalence of nasopharyngeal carriage of pneumococci in children varies between 51% and 97% in these regions, with the percentage of resistance to penicillin between 8.8% and 17% [79,80]. A prospective study of invasive pneumococcal infections in 2049 children hospitalized in Bamako with a fever of 39 °C or higher showed that 10% of deaths were related to pneumococcal infection, although only 2 of the 96 strains identified were not sensitive to penicillin [81]. It is clear that preventive penicillin remains necessary to prevent pneumococcal infections in sub-Saharan Africa. Similarly, pneumococcal vaccination is fully justified. The Pneumo23^®^ vaccine, effective after the age of 2, appears to be well suited for this purpose. By contrast, the effectiveness of Prevenar^®^ in Africa warrants further assessment, since a Malian study [81] indicated that serotype 5, which is not covered by Prevenar^®^, represented 54% of the strains that have been isolated. In contrast, a Gambian trial of a nine-valent conjugate vaccine reduced mortality by 16% among vaccinated children [82]. Other conjugated vaccines are being tested and are likely to be better suited to the African context than Prevenar^®^. Systematic prophylaxis by penicillin and anti-pneumococcal vaccination has reduced the risk of serious pneumococcal infections in all areas where the systematic use of this vaccine at birth has been adopted. In ATS, microbiological samples and serology’s are rarely positive: fibroscopy samples are most often taken after initiation of probabilistic antibiotics, and they are used more rarely in children than in adults [6]. In order to prevent bacterial infections, particular attention must be paid to hygiene, especially hand washing, which appears to be important for individuals with sickle cell anemia [83], while nutritional supplementation with zinc has been reported to be effective at reducing the risk of infection [84]. A small summary of the data from this study is shown in Table 1.

## 4. Conclusions

The organization of sickle cell care highlighting the prevention of severe bacterial infections, in particular pneumonia due to *S. pneumoniae* by systematic penicillin therapy and vaccinations as well as probabilistic antibiotic therapy protocols in the context of fever, has significantly improved the survival of sickle cell patients. The distribution of the bacterial strains responsible for these infections, which remains a major cause of morbidity and mortality in sickle cell anemia, particularly in low-income countries, varies from continent to country and from country to country, as well as sites of infection. The downside of antibiotic therapy has been the increased resistance of bacterial infectious agents to antibiotics.

Immune deficiency has often been discussed to explain the high frequency and severity of bacterial infections in sickle cell patients, but the burden of other risk factors promoting these infections, including biological and environmental, the bacterial ecology, and medical practices, must be studied within the framework of a “One Health” approach.

Thus, the control of the incidence and burden of bacterial infections in sickle cell populations by revising the guidelines for the prevention and management of bacterial infections in these patients could be improved. The key message from this work is summarized in Table 2.

## Figures and Tables

**Table 1 microorganisms-11-00859-t001:** The Contribution of this Study.

What This Study Adds?
Compiled available information concerning the high frequency of bacterial infections represents a major threat to public health in developing countries, where they remain responsible for significant morbidity and mortality in pediatric populations with sickle cell anemia;
Evidenced that the significantly high immunologic susceptibility is increased further for pneumococcal and salmonella infections in the population with sickle cell anemia;
Identified that some other factors contribute to promoting bacterial infections in individuals with sickle cell anemia, such as underdevelopment in some countries and socio-economic factors in a population with sickle cell anemia.

**Table 2 microorganisms-11-00859-t002:** Key Takeaway Message.

Key Messages
Several factors can be at the origin of a high susceptibility to bacterial infection in sickle cell patients: ➢ Immune deficiency, a factor that promotes pneumococcal and salmonella infections; ➢ Underdevelopment, a factor in the infection with pneumococcal and salmonella; ➢ Emergence of antibiotic resistance, a factor that favors bacterial infections. Systematic prophylaxis by penicillin and anti-pneumococcal vaccination has reduced the risk of serious pneumococcal infections.

## Data Availability

Not applicable.

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
