# Peer review of "Bacterial Infection in the Sickle Cell Population: Development and Enabling Factors"

_microorganisms, 2023, doi:10.3390/microorganisms11040859_

Round 1

Reviewer 1 Report

Thank you for the opportunity to review this manuscript. The research question on the factors associated with a bacterial infection in children with sickle cell disease is relevant and interesting. However, the manuscript has major methodological issues; there needs to be clarity on whether this is a narrative or a systematic review. I have highlighted some of the issues for your action before the manuscript is published,

Abstract:

1. Currently, it is more of a justification for the review; you can reduce the content and cut some words. There is no mention of the summary results (potential factors for bacterial infection). Please add the key findings (results) of your review.

2. Avoid lengthy lines, for example first line of the abstract. This comment applies to the whole of the manuscript.

Introduction:

1. Introduction is succinct and reads very well. However, it will be nice to highlight the current knowledge gaps and how this manuscript addresses them.

Methodology:

It is unclear whether this is a narrative or a systematic review. If you consider it a systematic review, you need to provide a detailed account of the methodology of review, especially on search strategy, data abstraction, selection of certain factors and so on. Therefore, it would be better to use the PRISMA checklist for drafting the methods section. On the other hand, if you consider it a narrative review (perspective), you can remove the whole section on Methods.  

Results:

It needs to be made clear how you picked these factors for detailed narration. Once there is clarity on the methodology, it will be easy to comment on the results.

Author Response

Abstract:

  1. Currently, it is more of a justification for the review; you can reduce the content and cut some words. There is no mention of the summary results (potential factors for bacterial infection). Please add the key findings (results) of your review. Applied 

2. Avoid lengthy lines, for example first line of the abstract. This comment applies to the whole of the manuscript.

Applied 

Introduction:

Introduction is succinct and reads very well. However, it will be nice to highlight the current knowledge gaps and how this manuscript addresses them.

Applied

Methodology:

It is unclear whether this is a narrative or a systematic review. If you consider it a systematic review, you need to provide a detailed account of the methodology of review, especially on search strategy, data abstraction, selection of certain factors and so on. Therefore, it would be better to use the PRISMA checklist for drafting the methods section. On the other hand, if you consider it a narrative review (perspective), you can remove the whole section on Methods.  

Applied 

Results:

It needs to be made clear how you picked these factors for detailed narration. Once there is clarity on the methodology, it will be easy to comment on the results.

Thank you 

Regards!

Reviewer 2 Report

My comments are limited to clarifications. 

Line 29: surely nutritional and maternal anaemia are more serious globally? Suggest 'form on inherited haemoglobin disease'.

Line 60-1: any more about developed countries?

Line 61-4: these two sentences mean the same thing? what is the point being made?

Line 82-3: ACP not APC

Line 99-100: I see no explanation for why big (sic) spleens cause functional asplenia

Line 104: the original article refers not to 'vacuolated' RBC but to 'vesiculated/pocked' RBC.

Line 125: what is PLS?

Line 296-7: fat embolism and marrow embolism are precipitators or ACS. Pulmonary artery thrombosis is the correct term.

Author Response

Line 29: surely nutritional and maternal anaemia are more serious globally? Suggest 'form on inherited haemoglobin disease'.

Applied

Line 60-1: any more about developed countries?

Throughout the manuscript the comparison between developed and developing countries is made. I wonder if developing this would be redundant for the reader!

Line 61-4: these two sentences mean the same thing? what is the point being made?

You are right indeed but it was to insist on the character of susceptibility to infection but I deleted one of the sentences.
Thank you for this remark 

Line 82-3: ACP not APC

Applied

Line 99-100: I see no explanation for why big (sic) spleens cause functional asplenia

frequent splenomegaly is a factor favouring pneumococcal infection in children with sickle cell disease.

Line 104: the original article refers not to 'vacuolated' RBC but to 'vesiculated/pocked' RBC.

Applied

Line 125: what is PLS?

Applied meaning

Line 296-7: fat embolism and marrow embolism are precipitators or ACS. Pulmonary artery thrombosis is the correct term.

the correction has been applied

Round 2

Reviewer 1 Report

Thank you for attending to all comments